# Development of a Simple Single-Acupoint Electroacupuncture Frame and Evaluation of the Acupuncture Effect in Rabbits

**DOI:** 10.3390/vetsci8100217

**Published:** 2021-10-05

**Authors:** Huan Huang, Jianrong Zhang, Fuxing Gui, Sheng Liu, Chonghua Zhong, Tingting Wang, Hongxu Du, Xianlin He, Liting Cao

**Affiliations:** 1Department of Traditional Chinese Veterinary Medicine, College of Veterinary Medicine, Southwest University, Rongchang District, Chongqing 402460, China; huangfuna128@email.swu.edu.cn (H.H.); zjianrong@email.swu.edu.cn (J.Z.); g17823734193@email.swu.edu.cn (F.G.); liu734483872@email.swu.edu.cn (S.L.); quyiwen@email.swu.edu.cn (T.W.); hongxudu@swu.edu.cn (H.D.); lwh810@email.swu.edu.cn (X.H.); 2Department of Animal Husbandry and Engineering, College of Animal Science and Technology, Southwest University, Rongchang District, Chongqing 402460, China; zhongchongh@swu.edu.cn; 3Chongqing Engineering Research Center of Veterinary Medicine, Rongchang District, Chongqing 402460, China; 4Chi Institute of Traditional Chinese Veterinary Medicine, Southwest University, Rongchang District, Chongqing 402460, China; 5Chongqing Three Gorges Vocational College, Wanzhou District, Chongqing 404155, China

**Keywords:** single-acupoint electroacupuncture, traditional electroacupuncture, ST-36 and/or LI-10, acupuncture effect

## Abstract

To reduce the circulation path of the output current of traditional electroacupuncture (TEA) process in the body, a simple single-acupoint electroacupuncture (SEA) frame was designed and the acupuncture effect of SEA was evaluated through Hou-san-li (ST-36) and Qian-san-li (LI-10) acupoints. Forty-two healthy New Zealand rabbits were randomly divided into seven groups and underwent acupuncture for 20 min in an awake state. Blood samples aseptically collected from the ear vein 3 h before acupuncture and 0, 3, 6, 9, 12 and 24 h after acupuncture were used for the detection of aspartate aminotransferase (AST), alanine aminotransferase (ALT), creatine kinase MB (CK-MB) and motilin (MTL) in serum. The simple SEA frame was developed successfully, and the acupuncture results showed that the serum AST and ALT levels were significantly higher at 3 h after TEA with high frequency (*p* < 0.01) compared with the control group. Regarding serum CK-MB levels, no significant differences were found after SEA or TEA stimulation (*p* > 0.05). Serum MTL levels were significantly increased at 0 h after SEA and TEA (*p* < 0.05), but there were no significant differences at other time points after SEA and TEA treatment (*p* > 0.05). SEA not only maintains the effect of TEA but also shortens the circulation loop of the electroacupuncture (EA) current in the body, which effectively avoids body injury.

## 1. Introduction

Acupuncture is a traditional therapy with a history of more than 3000 years and has been accepted worldwide as a complementary and alternative medicine. Acupuncture is based on the metaphysical concept of *Qi* [1], which is an important energy source of the body. The channels through which *Qi* circulates in the body are called meridians, so they can be used to balance Yin and Yang by regulating the flow of *Qi* through the stimulation of acupuncture or moxibustion [2]. In 1810, the French physician Louis Berlioz first proposed the idea of treating diseases with currents during acupuncture [3]. In this process, a needle is electrified with microcurrent waves of bioelectricity to stimulate acupoints with different currents and frequencies generated by microcurrent waves to promote the release of endogenous substances. Compared with traditional acupuncture, electroacupuncture is simpler and more replicable. Electropuncture is a popular complementary treatment option in humans and achieved some clinical results [4]. The output of the existing electroacupuncture instrument consists of at least two electrodes or two wires. In experimental research or clinical treatments, two or more different acupoints of the body should be connected, so that the electric current forms a circuit in the body to achieve the purpose of treatments. At present, electroacupuncture is widely applied in veterinary clinics [5,6]. It is well known that controlling the electrical current and accurately targeting acupoints directly affects the effectiveness of treatments. However, due to physiological differences among animal species, animals exhibit different degrees of sensitivity to electric currents. These differences in sensitivity cause some limitations in the clinical application of electroacupuncture in animals. Many species of animals have relatively small bodies, such as rabbits and mice, and are more sensitive to electrical currents than other animals. Therefore, there is great potential for electric currents to damage the body during electroacupuncture therapy in these animals [7]. How to choose the right acupoint in animals and reduce damage to the body caused by currents to improve the safety of electroacupuncture treatment is a key issue in electroacupuncture treatment.

Therefore, based on the characteristics of traditional needling therapy and the advantages of traditional electroacupuncture (TEA), this study was conducted to design a simple single-acupoint electroacupuncture (SEA) frame. This electroacupuncture frame allows an electrical current generated by positive and negative electrodes to form a loop in the acupuncture point and simultaneously expand the local stimulation range. To verify the effectiveness and safety of this electroacupuncture frame, the effects of electrical current on biochemical indicators in New Zealand rabbits, which are sensitive to electrical stimulation, were observed after electroacupuncture stimulation of Qian-san-li (LI-10) and Hou-san-li (ST-36) acupoints at different frequencies. This study will provide a new concept for the wide application of electroacupuncture in veterinary clinics, especially in the prevention and treatment of small animal diseases.

## 2. Materials and Methods

### 2.1. Animals

Forty-two healthy male New Zealand rabbits weighing 2.0 ± 0.4 kg were used in the study and were purchased from the Chongqing Academy of Chinese Materia Medica (Chongqing, China). The rabbits were housed in clean cages at a temperature of 24 ± 2 °C and relative humidity of 55 ± 15%. All animal experiments and procedures were approved by the Laboratory Animal Ethical Commission of Southwest University (Chongqing, China).

### 2.2. Ideas for the Development of a Simple SEA Frame

The new simple SEA frame was created using traditional acupuncture needles and copper needle holders, as shown in Figure 1. The holder consisted of four legs, and the metal ring had an insulating layer of polypropylene resin. The legs linked to the metal ring were evenly distributed under the metal ring. An electric needle fixation device, which consisted of a cross-designed movable buckle, was also provided to fix the electric needle. Slip rings that held the metal ring and slide along it were attached to both ends of the movable buckle. A small triangle formed by crossing two movable buckles was used to fix the electric needle. By sliding the slip ring, the fixed position could be randomly changed.

### 2.3. Safety Evaluation and Effect Verification of Simple SEA

Forty-two healthy New Zealand rabbits were randomly divided into 7 groups of 6 animals each: the control group (group C), TEA with high frequency group (acupuncture at ST-36 and LI-10 together) (group TEA-HF), TEA with low frequency group (acupuncture at ST-36 and LI-10 together) (group TEA-LF), SEA with high frequency group I (right ST-36) (group SEA-HFI), SEA with low frequency group I (right ST-36) (group SEA-LFI), SEA with high frequency group II (acupuncture at ST-36 and LI-10,respectively) (group SEA-HFII) and SEA with low frequency group II (acupuncture at ST-36 and LI-10,respectively) (Group SEA-LFII). The control group underwent acupuncture without receiving current, and the other groups received electroacupuncture. The parameters of electroacupuncture stimulation were as follows: the frequency of low-frequency electroacupuncture was 2 Hz, and that of high-frequency acupuncture was 100 Hz; the duration was 20 min; the waveform was continuous; and needle depth was 7 mm. After −3, 0, 3, 6, 9, 12 and 24 h, 1.5 mL of auricular venous blood was collected, then placed in EP tubes for 1–2 h, and centrifuged at 3000 rpm for 20 min to separate the serum. The separated serum was put into EP tubes, numbered and stored in a refrigerator at −80 °C.

### 2.4. Serum Biochemical Indexes

Aspartate aminotransferase (AST) and alanine aminotransferase (ALT) from all the experimental serum samples collected were measured with assay kits from Nanjing Jiancheng Bioengineering Institute (Jiangsu, China) according to the manufacturer’s instructions. In brief, 50 µL of serum sample was mixed with 250 µL of matrix solution and incubated at 37 °C for 30 min. Next, 250 µL of chromogenic solution was applied to the wells and incubated at 37 °C for 20 min. Finally, 1:9 dilution of termination solution was applied to the wells and the optical density (OD) values were read at 505 nm with G9 double beam UV-visible spectrophotometer from Nanjing Philes (Jiangsu, China).

Creatine kinase MB (CK-MB) and motilin (MTL) levels were similarly measured with the enzyme-linked immunosorbent assay (ELISA) kits bought from Nanjing Jiancheng Bioengineering Institute (Jiangsu, China) according to the manufacturer’s instructions. ELISA plates were recovered to room temperature before testing. Briefly, 50 µL of serum samples were mixed with 50 µL of biotin antigen and incubated for 30 min at 37 °C. After that, 50 µL of HRP-avidin was applied to the wells after washing 4 times with PBS and incubated for 30 min at 37 °C. After washing, 50 µL of chromogenic solution A and B were added sequentially and incubated for 10min. Finally, the reaction was stopped by adding 2 M sulfuric acid and the optical density (OD) values were read at 450 nm with a microplate reader from Fuan (Shanghai) Enterprise (Shanghai, China).

### 2.5. Statistical Analysis

The data were analyzed using SPSS 19.0 statistical software. The data are expressed as the mean ± standard deviation ($\bar{X}\;\pm \;SD$). Single factor analysis of variance was used to compare the means of multiple groups, and the Least Significant Difference (LSD) method was used to compare those of two groups. Differences were considered significant if *p* < 0.05.

## 3. Results

### 3.1. Safety Evaluation of the Simple SEA Frame

The acupoints were selected dialectically, and routine acupuncture was performed. After the desired parameters were set, an animal was placed in the simple SEA frame and fixed with the canine Baoding frame. As shown in Figure 2, the positive and negative electrodes were connected to a bracket in the needle handle and electroacupuncture frame, respectively. The electroacupuncture machine was turned on, and the waveform and frequency were adjusted to the appropriate range. In addition, the reaction of the animals was also observed. After acupuncture and electrification, at the front of New Zealand rabbit appeared a slight tremble, a slight sound, individual defecation and other phenomena. Ten minutes after electroacupuncture, on the acupuncture side of New Zealand rabbit appeared a slight tremble, limb lifting and other sound phenomenon. Eighteen minutes after acupuncture, the New Zealand rabbits appeared to have a concave waist and muscle tremor, and some rabbits appeared humming and limb lifting, showing a state of “*D**e Qi*”.

### 3.2. Effects of SEA on Serum AST Levels in Rabbits

As shown in Table 1, the AST level in group TEA-HF at 3 h after electroacupuncture was extremely significantly different from that in control group (*p < 0.01*), but no significant difference was noted among the other groups.

### 3.3. Effects of SEA on Serum ALT Levels in Rabbits

As shown in Table 2, the ALT levels in group TEA-HF at 3 h after electroacupuncture were extremely significant difference compared with control group (*p < 0.01*), but no significant difference was noted among the other groups.

### 3.4. Effects of SEA on Serum CK-MB in Rabbits

As presented in Table 3, there was no significant difference in serum CK-MB levels between the groups that received simple single-point electroacupuncture and those that received traditional electroacupuncture (*p* > 0.05).

### 3.5. Effects of SEA on Serum MTL Levels in Rabbits

As shown in Table 4, the MTL levels in groups TEA-HF, SEA-HFI and SEA-HFII at 0 h after electroacupuncture was significantly different from that in the control group (*p* < 0.05). The MTL content in each group at 0 h after electroacupuncture was significantly different from that at 3 h before electroacupuncture (*p* < 0.05).

## 4. Discussion

“*De Qi*” is a state of internal *Qi* and blood changes after meridians are stimulated. When deep acupuncture at acupoints is performed, needling sensations will occur, such as acid, numbness, distension, pain and swelling [8], or a mixture of one or more sensations. The appearance of “*De Qi*” can improve the imbalance of the body and exert a therapeutic effect [9]. However, in electroacupuncture, biological waves replace needle movement by the operator after acupuncture to achieve “*De Qi*”, thus simplifying and standardizing the influence of different manual acupuncture operations. However, given the limited communication ability of animals, they are unable to describe the response to electroacupuncture as humans can. Moreover, it is very difficult to judge whether *Qi* is generated in animals due to the damage caused by the stimulation of electric current to the animal body, or whether *Qi* is generated in animals and is directly affected by the therapeutic effect of electroacupuncture. Scholars have used magnetic resonance imaging (MRI) to study the neuronal specificity of acupuncture points and found that deep acupuncture of meridians and collaterals after “*De Qi*” yielded a stronger acupuncture sensation than deep acupuncture and subcutaneous acupuncture without meridians; and that it had a more effective regulatory effect on the pain-related nerve matrix [10]. There are many different acupuncture techniques reported in Huangdi Neijing (The Yellow Emperor’s Inner Classic), with differences in the structure and size range of acupoints. However, the emergence of new acupoint fixation methods, such as three-dimensional acupoint fixation with nuclear magnetic resonance (NMR) [11], has also indicated that acupoints may be regions rather than points, namely, “acupoint areas”. In addition, stimulating these areas can obviously induce similar therapeutic effects under pathological conditions. Theoretically speaking, acupoints have different shapes and should not be limited to points [12]. In this study, a four-legged simple SEA frame, which can provide a reasonable stimulation point/range on the basis of traditional acupuncture, was developed. The effectiveness of single-point TEA was determined, which is more conducive of the “*De Qi*” reaction.

Appropriate acupuncture points need to be dialectically selected for traditional acupuncture therapy, and these acupuncture points need to be stimulated in a certain order through lifting, inserting, twisting, turning and other acupuncture techniques. Since the popularization of electroacupuncture, some scholars have found that the analgesic effects of electroacupuncture are prominent. Further studies have revealed that different frequencies result in the production of different molecules in the central nervous system. Specifically, 2-Hz electroacupuncture promotes the release of enkephalins, β-endorphins and endorphins, while 100-Hz electroacupuncture selectively increases the release of dynorphin [13]. All opioid peptides are released in a combination of two frequencies for a maximum therapeutic effect [14]. Although electroacupuncture has been described as a safe alternative to therapy, there are certain risks associated with the treatment. A previous study found that electroacupuncture can change the local microvascular circulation near the cardiac end [15]. When electroacupuncture is used to stimulate the body and the needle is placed at an adjacent acupoint on one side of the limb, there is a little current that flows along the limb into the chest. However, when a pair of needles are placed on the opposite elbow or below the knee, the electric field generated by the needles can create an electric current in the chest cavity and affect the heart; thus, current should not be allowed to pass through the heart during electroacupuncture therapy [16]. Electroacupuncture has been shown to be effective in the treatment of central neuralgia and visceral pain in animals. However, because small animals, such as rabbits and mice, are smaller and are more sensitive to electricity, these animals are less able to be calmed with tactile or vocal reassurance than domestic cats and dogs. In addition, many of our animal companions are actually prey and alert to unfamiliar sensations and environmental conditions. Birds and small mammals (rabbits, guinea pigs, chinchillas) are less tolerant to electric current, but large animals such as cows and tigers are more tolerant to electricity. The anatomical diversity among species and the resulting difficulty in point shifting increase the difficulty of performing electroacupuncture in animals [17]. The lack of a gold standard for pain assessment and differences in the selection of currents and frequency seriously hinder our ability to perform acupuncture on companion animals, zoo animals and wild animals [18]. Therefore, a simple SEA frame was developed, and the effects of simple SEA and traditional acupuncture on healthy New Zealand rabbits at the Hou-san-li (ST-36) and/or Qian-san-li (LI-10) acupoints were compared by measuring the levels of organ injury-related hormones to assess the influence of different frequencies on the body.

AST is mainly distributed in myocardium, liver and skeletal muscle. Within these cell types, it exists in mitochondria and cytoplasm, and when hepatocyte injury is found, for example, it can lead to the release of ALT in mitochondria, resulting in an increase in its level in hepatocytes. When liver cells are seriously damaged, the mitochondria of liver cells are further damaged, and the liver releases a large amount of ALT into the blood, resulting in a significant increase in ALT levels in the blood [19]. Compared with those in the control group, AST levels in the TEA-HF group were significantly increased, but no significant difference was noted in the other experimental groups. Similar results were found for serum ALT levels. AST/ALT ratio is also often used as a reference for liver function injury in combination with AST and ALT. In New Zealand rabbits with liver injury, the value is often increased at the same time as AST and ALT. When the AST/ALT ratio is greater than 1.6, it indicates that the liver injury is serious [20].

CK-MB is an important indicator for the diagnosis of myocardial injury. CK-MB is an isoenzyme of creatine kinase (CK), which mainly exists in myocardial tissue. When myocardial cells die, a large number of isoenzymes are released into the blood with acute myocardial infarction [21]. According to the results of this study, the CK-MB contents after simple SEA and TEA were in the normal range, which suggests that simple SEA and TEA have similar effects when administered at the two acupoints of the Yang Ming meridian. It is also worth studying whether they have an effect on the function of meridian acupoints when administered at two acupoints not on the same meridian.

The Hou-san-li acupoint is the compound point of the Yang Ming stomach meridian, and stimulation of this acupoint has a good therapeutic effect in protecting the gastrointestinal mucosa [22,23]. Clinical studies have shown that electroacupuncture stimulation of the Hou-san-li acupoint may restore gastric antrum movement caused by rectal dilatation by enhancing vagus nerve activity in dogs, which is partly mediated by the opioid pathway [24]. Some studies have found that the concentrations of gastrointestinal-related hormones, such as vasoactive intestinal peptide, motilin and gastrin, are increased in mice subjected to acupuncture at the Hou-san-li acupoint compared with those not subjected to acupuncture [25]. MTL is a straight-chain polypeptide composed of 22 amino acids first described in 1966 by J C Brown. It is secreted by Mo cells and released periodically during digestion. Motilin can induce strong gastric contraction and segmented movement of the small intestine, and the decrease of its secretion can lead to the weakening of gastrointestinal contraction, so its physiological function is mainly to promote gastrointestinal motility and gastric emptying [26,27,28]. The effects of simple SEA and TEA on motilin levels in New Zealand rabbits were observed after acupuncture at the Hou-san-li acupoint. The results showed that serum MTL levels in rabbits were significantly increased at the end of electroacupuncture, and the change in MTL content after electroacupuncture was greater than that during electroacupuncture. This result is similar to Lin’s result indicating that acupuncture at the Hou-san-li acupoint can enhance gastrointestinal myoelectric activity and MTL levels of New Zealand rabbits under normal physiological conditions [29]. Yu noted that relevant clinical and experimental studies have reported inconsistent findings related to the effects of electroacupuncture at different frequencies on the digestive and other systems, indicating that the effects of electroacupuncture are unclear; thus, appropriate electroacupuncture parameters remain to be determined [30]. The etiology, location and nature of clinical diseases are also different. For the application of acupuncture treatment, it is necessary to select acupoints according to the specific manifestations of the disease or condition while following the principle of treating diseases according to their unique characteristics and presentation.

## 5. Conclusions

We have successfully developed a simple SEA frame to shorten the circulation loop of the EA current in the body. And we have demonstrated that this simple SEA frame for electroacupuncture is safe and effective by detection of AST, ALT, CK-MB and MTL in serum of New Zealand rabbits after LI-10 and ST-36 acupoints stimulation. Finally, this device can be used as a safe and effective alternative to TEA prevention and treatment for animal diseases.

## Figures and Tables

**Figure 1 vetsci-08-00217-f001:**
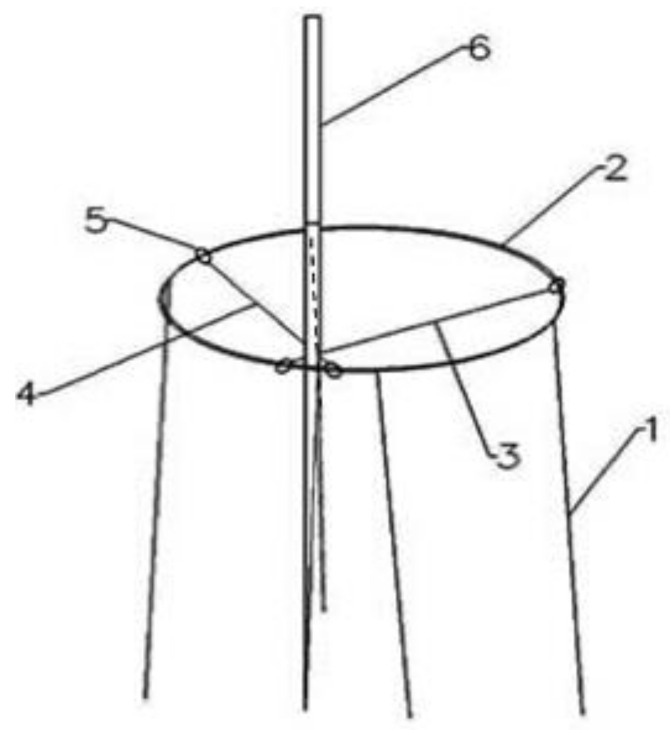
Simple single-acupoint electroacupuncture (SEA) design. (1) bracket, (2) provided with an insulating layer on the outer edge of the metal ring, (3) and (4) movable buckles, (5) slip ring of the movable buckle, and (6) acupuncture needle.

**Figure 2 vetsci-08-00217-f002:**
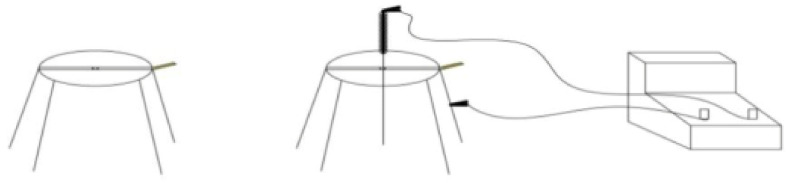
Drawing of the simple single-point electroacupuncture (SEA) frame. From left to right are the electroacupuncture rack, single point electroacupuncture frame and electroacupuncture apparatus.

**Table 1 vetsci-08-00217-t001:** Effects of different treatment methods on AST activity at different time points (*U/L, n = 6*).

Group	−3 h	0 h	3 h	6 h	9 h	12 h	24 h
C	16.58 ± 1.57	17.00 ± 1.87	17.97 ± 0.73	16.72 ± 0.71	17.41 ± 1.71	17.19 ± 1.10	17.15 ± 1.10
TEA-HF	17.27 ± 1.23	19.08 ± 0.70	32.83 ± 0.89 **	18.52 ± 0.66	18.39 ± 0.66	18.03 ± 0.20	18.03 ± 0.64
TEA-LF	17.69 ± 1.55	18.13 ± 0.53	23.66 ± 0.79	18.10 ± 1.13	19.16 ± 0.30	18.06 ± 0.46	17.19 ± 0.58
SEA-HFI	15.81 ± 0.66	18.58 ± 0.75	23.62 ± 0.87	17.65 ± 1.08	17.99 ± 1.17	18.33 ± 0.29	17.45 ± 0.68
SEA-LFI	15.18 ± 0.91	16.87 ± 0.86	23.03 ± 0.89	16.59 ± 1.38	15.90 ± 1.11	16.05 ± 0.75	15.37 ± 0.85
SEA-HFII	17.83 ± 1.06	18.60 ± 0.43	24.82 ± 0.61	18.51 ± 0.75	18.82 ± 0.83	18.79 ± 0.91	18.24 ± 0.45
SEA-LFII	15.60 ± 0.81	17.22 ± 0.37	23.57 ± 0.47	19.06 ± 0.31	17.36 ± 1.88	17.16 ± 0.89	15.99 ± 0.33

** indicates an extremely significant difference (*p < 0.01*).

**Table 2 vetsci-08-00217-t002:** Effects of different treatment methods on ALT activity at different time points (*U/L, n = 6*).

Group	−3 h	0 h	3 h	6 h	9 h	12 h	24 h
C	22.15 ± 0.03	23.62 ± 0.70	25.83 ± 0.83	25.71 ± 1.03	25.83 ± 0.72	22.17 ± 0.04	22.12 ± 0.18
TEA-HF	23.21 ± 0.72	24.02 ± 0.56	57.20 ± 1.30 **	27.94 ± 0.63	27.50 ± 0.44	27.10 ± 91.0	25.73 ± 0.47
TEA-LF	21.56 ± 2.62	25.13 ± 0.84	21.85 ± 0.25	25.11 ± 0.17	24.58 ± 0.79	25.19 ± 0.76	24.96 ± 0.50
SEA-HFI	21.62 ± 0.95	24.41 ± 0.70	29.56 ± 0.95	25.94 ± 0.24	25.00 ± 0.72	23.74 ± 1.03	22.60 ± 0.84
SEA-LFI	22.08 ± 0.48	25.71 ± 0.26	26.14 ± 1.49	25.16 ± 0.53	24.42 ± 0.30	24.11 ± 0.49	23.11 ± 0.74
SEA-HFII	22.77 ± 0.81	22.88 ± 0.80	20.69 ± 0.17	26.84 ± 1.10	24.53 ± 1.30	23.26 ± 0.70	22.33 ± 0.54
SEA-LFII	20.83 ± 0.04	22.27 ± 0.94	26.02 ± 0.73	26.40 ± 1.72	23.44 ± 0.45	22.12 ± 0.85	20.09 ± 0.04

** indicates an extremely significant difference (*p* < 0.01).

**Table 3 vetsci-08-00217-t003:** Effects of different treatment methods on CK-MB activity at different time points (*U/L, n = 6*).

Group	–3 h	0 h	3 h	6 h	9 h	12 h	24 h
C	7.61 ± 3.00	10.21 ± 1.96	11.65 ± 3.82	7.81 ± 1.42	9.37 ± 3.28	9.37 ± 1.57	8.35 ± 1.88
TEA-HF	8.32 ± 2.54	9.08 ± 3.44	10.21 ± 3.65	8.33 ± 3.97	7.18 ± 3.45	10.44 ± 3.64	10.60 ± 4.38
TEA-LF	7.90 ± 2.75	9.68 ± 3.00	8.49 ± 3.47	9.28 ± 4.15	8.41 ± 3.88	8.76 ± 3.67	11.63 ± 3.23
SEA-HFI	10.93 ± 2.24	7.85 ± 3.48	7.52 ± 4.80	8.23 ± 2.58	10.44 ± 3.52	6.59 ± 3.59	9.89 ± 3.71
SEA-LFI	9.94 ± 4.01	11.00 ± 2.83	10.57 ± 1.90	8.38 ± 1.80	11.37 ± 2.55	9.32 ± 3.95	7.89 ± 1.80
SEA-HFII	10.52 ± 2.10	8.16 ± 4.15	8.89 ± 3.77	7.33 ± 3.44	12.95 ± 0.99	9.20 ± 3.40	10.86 ± 4.31
SEA-LFII	10.57 ± 2.47	9.02 ± 3.32	9.90 ± 3.65	7.19 ± 3.50	8.90 ± 3.90	9.77 ± 3.05	7.40 ± 3.76

**Table 4 vetsci-08-00217-t004:** Effects of different treatment methods on MTL activity at different time points (*ng/L, n = 6*).

Group	–3 h	0 h	3 h	6 h	9 h	12 h	24 h
C	131.46 ± 7.90	145.66 ± 17.15	110.65 ± 13.25	111.02 ± 14.95	116.32 ± 22.82	124.76 ± 13.97	109.95 ± 13.70
TEA-HF	126.04 ± 16.72	272.21 ± 45.66 *^,#^	129.53 ± 13.44	107.08 ± 16.49	117.37 ± 18.88	112.79 ± 18.29	110.63 ± 15.79
TEA-LF	124.96 ± 20.55	193.02 ± 26.26 ^#^	125.28 ± 16.35	119.29 ± 15.13	117.98 ± 18.77	104.30 ± 13.56	113.27 ± 15.54
SEA-HFI	122.67 ± 19.52	262.40 ± 59.44 *^,#^	120.38 ± 23.12	118.65 ± 12.80	110.21 ± 17.09	114.80 ± 12.11	118.43 ± 14.61
SEA-LFI	123.74 ± 21.50	179.72 ± 23.06 ^#^	111.64 ± 15.19	116.33 ± 15.73	115.28 ± 16.16	117.48 ± 19.65	124.72 ± 12.97
SEA-HFII	135.41 ± 8.78	278.19 ± 39.87 *^,#^	131.71 ± 12.15	126.99 ± 20.98	115.18 ± 19.51	124.55 ± 19.73	123.37 ± 15.49
SEA-LFII	122.06 ± 17.91	181.11 ± 29.98 ^#^	115.46 ± 12.25	120.42 ± 14.42	120.08 ± 14.21	121.82 ± 10.79	115.23 ± 14.50

Note: * indicates a significant difference compared with control group (*p* < 0.05), ^#^ indicates compared with 3 h before electroacupuncture (*p* < 0.05).

## Data Availability

All data analyzed during this study are included in this published article.

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
