# Peer review of "Development of a Simple Single-Acupoint Electroacupuncture Frame and Evaluation of the Acupuncture Effect in Rabbits"

_vetsci, 2021, doi:10.3390/vetsci8100217_

Round 1

Reviewer 1 Report

MDPI Manuscript Review for: Vetsci-1373129

This is an interesting “safety” study of single acupoint electroacupuncture (SEA) using a new device. Although the authors have not performed an efficacy study to compare traditional electroacupuncture (TEA) with SEA, it is an important first step toward avoiding potential damaging effects to organs, especially liver and heart, of TEA. The MTL data is also encouraging about potential equivalent efficacy of SEA compared to TEA, although more formal studies of safety and efficacy of SEA need to be performed in future.

Specific comments on the manuscript are below:  

Line 22: Replace “CKMB and MTL in serum;” with “CKMB and MTL in serum.”

Line 23: Replace “serum AST and ALT levels were extremely significantly higher” with “serum AST and ALT levels were significantly higher”

Line 24: Replace “with high frequency (P<0.01) compared with control group” with “with high frequency (P<0.01) compared with the control group”

Line 21-22: “AST, ALT, CKMB and MTL in serum” All of these abbreviations must be written out in full at their first appearance, so here in the Abstract. Any time a new abbreviation appears, it must be spelled out in full and then the abbreviation follows in parentheses; then after that you can use just the abbreviation.

Line 26-27: Replace “but no significant differences on SEA and TEA (P>0.05)” with

 “but no significant differences between SEA and TEA (P>0.05) were observed.”

Line 27-28: “SEA not only maintains the effect of TEA but also shortens the circulation loop of the EA current in the body, which effectively avoids body injury.” You must specify what is meant by the “circulation loop” and provide an operational definition of what this means. Assume that people reading this article are not experts in electronics.

Line 35: Replace “The channels through which Qi circulates in the body is called” with “The channels through which Qi circulates in the body are called”

Line 42-43: “Electropuncture has good clinical efficacy in both humans and animals.” This is an overstatement and must be made more precise. For what conditions exactly has acupuncture been shown through rigorous clinical trials to have efficacy on an international level? Please site a Cochrane Review or other well-respected systematic review of clinical trials in acupuncture for specific medical condition (i.e., https://pubmed.ncbi.nlm.nih.gov/16734078/) or ( https://www.cochranelibrary.com/cdsr/doi/10.1002/14651858.CD013814/full?cookiesEnabled). The authors must make sure to cite recent and up-to-date literature as references and not references that are decades old.

Line 45-49: This is well stated and the reviewer commends the authors for this thoughtful statement about physiological differences among species and reaction to electrical current stimulation

Line 57-58: Replace “This electroacupuncture frame allows an electrical current generated by positive and negative electrodes form a loop in the acupuncture point” with “This electroacupuncture frame allows an electrical current generated by positive and negative electrodes to form a loop in the acupuncture point.”

Line 60: Replace “which are sensitive to electrical stimulate” with “which are sensitive to electrical stimulation,”

Line 62: Replace “This study provides a new idea” with “This study provides a new concept”

Line 68: Replace “Chongqing Academy of Chinese Materia Medical” with “Chongqing Academy of Chinese Materia Medica”

Line 89: Replace “(acupuncture at ST-36 and LI-10 respectively” with “(acupuncture at ST-36 and LI-10, respectively”

Line 90: Replace “(acupuncture at ST-36 and LI-10 respectively” with “(acupuncture at ST-36 and LI-10, respectively”

Line 99: “3000 rpm/min to separate the serum” Please state how long the samples were spun, for example 3000rpm for 5 min.

Line 100-102: “After all serum samples 100 were collected, AST, ALT, CKMB and MTL levels were measured with ELISA kits accord-101 ing to the kit instructions

Line 104-105: “The data are expressed as the mean ± standard deviation (?Ì…±???).” (?Ì…±???) this term refers to Standard Error of the Mean, and not the standard deviation, as stated above. Please correct this statement.

Lines 104-107: please make sure an experienced statistician has looked at your statistical analysis to make sure it is sound and correct.

Line 106: “and the LSD method was used to compare those” Please state what LSD is before using the abbreviation. Least Significant Difference (LSD).

Line 112: Replace “As shown in Figure 2, The positive” with “As shown in Figure 2, the positive”

Line 110-112: “The acupoints were selected dialectically, and routine acupuncture was performed. After the desired parameters were set, an animal was placed in the simple SEA frame and fixed.” Please describe in more detail how the New Zealand rabbit were “fixed” into place. How were they restrained? How stressful was this for them? Did they show strong resistance? Was any anesthesia used that would affect results in order to calm the animals down during restrain? This could have important implications for the results, so must be discussed in detail. It would be better to show a diagram also of the New Zealand rabbits in the SEA frame, and this would help the reviewer understand how the animal was placed and restrained and acupuncture “points” used.

Tables I and II: The observation of a significant rise in AST and ALT at the 3 hour timepoint is very interesting, but it is completely unclear where the organ injury is occurring, or perhaps across multiple organs.

Table 4: What does the symbol “#” represent? Please specify in a Table 4 footer below Table 4.

Line 128-129:and ** 128 indicates an extremely significant difference.” Please define specifically what extremely significant difference is. I am assuming it is p < 0.01.

Line 142-143: First, please define MTL as this reviewer does not know what biomarker you are referring to here. Are you referring to Motilin (MTL) and is this a measure of liver damage?

Line 142-143:  Replace “As shown in Table 4, the MTL levels in groups DH, SH1 and SH2 at 0 h after electroacupuncture was significantly different from that in the blank group (P < 0.05).” with “As shown in Table 4, the MTL levels in groups DH, SH1 and SH2 at 0 h after electroacupuncture was significantly different from that in the control group (P < 0.05).

Line 150-151: Replace “which may appear as a mixture of one or 150 sensations, occur.” with “which may appear as a mixture of one or more sensations, occur.”

Line 162: Replace “deep acupuncture and subcutaneous acupuncture without meridians” with “deep acupuncture and subcutaneous acupuncture without meridians;”

Line 173-174” Replace “The ineffectiveness of single-point TEA will be determined, which is more conducive to the “De Qi” reaction.” with “The effectiveness of single-point TEA will be determined, which is more conducive of the “De Qi” reaction.”

Line 176: Replace “and these acupuncture points needed to be stimulated in a certain order” with “and these acupuncture points need to be stimulated in a certain order”

Line 182-183: “while 100-Hz electroacupuncture selectively increases the release of strong enkephalins” Please state exactly which enkaphalins are released with 100Hz. Strong enkaphalins is meaningless.

Line 184: “Although electroacupuncture has been described as 184 a safe alternative to therapy,” What do you mean by therapy? What type of therapy and for what condition specifically?

Line 196: Replace “many of our animal companions are prey” with “many of our animal companions are actually prey”  

Line 203: Replace “on companion animals, zoo animals, and wild animals” with “on companion animals, zoo animals and wild animals”

Line 204-207: Replace “Therefore, a simple SEA frame was developed, and the effects of simple SEA and traditional acupuncture rabbit at the Hou-san-li acupoint were compared by measuring the 205 levels of organ injury-related hormones to assess the influence of different frequencies on the body.” with “Therefore, a simple SEA frame was developed, and the effects of simple SEA and traditional acupuncture on healthy New Zealand rabbits at the Hou-san-li acupoint were compared by measuring the levels of organ injury-related hormones to assess the influence of different frequencies on the body.

Line 208-209: “AST is the most abundant substance in the heart and is a sensitive indicator of myocardial cell injury.” This statement is not true and if it were should have been referenced. AST/ALT ratios are commonly used to indicate liver status and/or injury to the liver, not injury to the heart. Please remove this statement or justify with references. Aspartate aminotransferase (AST), like alanine aminotransferase (ALT) is found IN A VARIETY of tissues, not specific to heart muscle cells. Please review and update this entire paragraph for medical scientific accuracy with appropriate referencing (https://www.sciencedirect.com/topics/medicine-and-dentistry/aspartate-aminotransferase).

Line 209-210: “There is a very close relationship between the release of AST from the heart into the blood and myocardial injury.” Again, AST is found in a variety of tissues, not just heart, although there may be enrichment of this enzyme in heart AND OTHER tissues such as liver and skeletal muscle (https://www.sciencedirect.com/topics/medicine-and-dentistry/aspartate-aminotransferase). It is also highly enriched in liver and skeletal muscle, and the AST/ALT ratio is used to assess liver injury versus muscle injury. The most sensitive markers of cardiomyocyte injury are Troponin T (TnT) and Troponin I (TnI), along with creatine kinase MB (CK-MB). Because AST is found in many other tissues, including enrichment in liver, it is no longer used as a sensitive biochemical marker for cardiomyocyte injury (https://www.ncbi.nlm.nih.gov/pmc/articles/PMC4975226/).

Line 212-213: Replace “resulting in a significant increase in AST levels in the blood” with “resulting in a significant increase in ALT levels in the blood.” Please note the AST/ALT ratio is used to try to help determine whether injury is to the liver or to skeletal muscle. Again, please consult with an expert to make sure your statements are current and accurate, and provide up-to-date references.

Line 213-215: Replace “Compared with those in the control group, AST levels in the DH group were significantly increased, but no significant difference was noted in other tissues” with “Compared with those in the control group, AST levels in the DH group were significantly increased, but no significant difference was noted in the other experimental groups.”

Line 220-221: Replace “a large number of isoenzymes are released into the blood to induce acute myocardial infarction” with “a large number of isoenzymes are released into the blood with acute myocardial infarction”

Line 215-217: Replace “Similar results were found for serum ALT levels, considering that traditional double-point electroacupuncture may cause some damage to the viscera under high-frequency stimulation.” With “Similar results were found for serum ALT levels.”

Line 238-240: Replace “and the change in MTL content change after electroacupuncture was greater than that during electroacupuncture.” with “and the change in MTL content after electroacupuncture was greater than that during electroacupuncture.”

Line 240-241: Replace “This result is similar to Lin's result indicating that acupuncture at Hou-san-li acupoint” with “This result is similar to Lin's result indicating that acupuncture at the Hou-240 san-li acupoint”

Line 247-249: Replace “it is necessary to 247 select acupoints according to the specific conditions of the disease and differential treatment of syndromes and follow the principle of treating diseases according to the situation.” with “it is necessary to select acupoints according to the specific manifestations of the disease or condition while following the principle of treating diseases according to their unique characteristics and presentation.”

Line 251: Replace “by ensuring the effect of traditional acupuncture, overcomes” with “by ensuring the effect of traditional acupuncture overcomes”

Line 253-254: Replace “Moreover, the effects of local current formation and its mechanism need to be further studied.” with “The effects of local current formation and its mechanism need to be further studied.”

Reviewer 2 Report

The authors present an interesting study examining the potential of a custom frame for applying electroacupuncture to small animals. Briefly, the authors explore a number of conditions and parameters of electric current to a number of experimental groups of rabbits, with blood draws taken at various time points allowing insight into the release of biomarkers of injury of several organ systems. Taken together, this article gives insight into the potential of the aforementioned frame for electroacupuncture, towards improving delivery of the therapy to small animals to the benefit of the user, and the recipient.

In reviewing the manuscript however I had a number of concerns. The following should be addressed when preparing a suitable revision.

  1. Why did the authors not include a group of animals that did not receive acupuncture? It would have been useful to see whether acupuncture in and of itself had any impact on the baseline levels of the biomarkers examined.
  2. Information pertaining to the behaviour of the animals in receiving the treatment, and post-treatment, should be documented. Signs of stress, markings to the animals, etc should all be noted.
  3. Information in the table legend as to what the symbol for significance stands for should be given. For example, is the result significant to a specific time point, or a specific experimental group?
  4. More information on how the biomarkers of interest were measured should be given. IT is not enough to simply mention ELISA, and information on the exact kits, and how they were performed should be detailed.
  5. Can the authors confirm if the units for the table are ‘units per litre’? Typically an ELISA will measure markers of interest in units such as pg/ml or similar. Clarity on this should be addressed.
  6. There are several abbreviations used throughout which are not expanded upon. These should be expanded upon in the first instance such as to inform the reader as to what exactly they are.
  7. The gender of the animals does not appear to be listed. This should be clarified.

Round 2

Reviewer 2 Report

The authors have responded favorably to my comments and in their response the manuscript is much improved.

However I still have a number of concerns. In clarifying that the units for the biomarkers is indeed in units per litre this appears to be incredibly low for each of the biomarkers, in particular all of them with the exception of perhaps MTL. Did the authors include blank well or negative controls in the experiment? How did the signal measured compare to the values obtained? I have doubts that the biomarker in question was measured/detectable and instead some of these values could be background noise.  
